# Modeling and comparing data mining algorithms for prediction of recurrence of breast cancer

**Alireza Mosayebi**[1]☯, **Barat Mojaradi**[2]☯*, **Ali Bonyadi Naeini**[1]☯, **Seyed Hamid Khodadad Hosseini**[3]☯

**1** Department of Management and Business Engineering, School of Progress Engineering, Iran University of Science and Technology, Tehran, Iran, **2** Department of Geomatics, School of Civil Engineering, Iran University of Science and Technology, Tehran, Iran, **3** Department of Management, Faculty of Management and Economics, Tarbiat Modares University, Tehran, Iran

☯ These authors contributed equally to this work.
* mojaradi@iust.ac.ir

**Data Availability Statement:** All relevant data are within the manuscript and its Supporting Information files.

## Abstract

Breast cancer is the most common invasive cancer and the second leading cause of cancer death in women. and regrettably, this rate is increasing every year. One of the aspects of all cancers, including breast cancer, is the recurrence of the disease, which causes painful consequences to the patients. Moreover, the practical application of data mining in the field of breast cancer can help to provide some necessary information and knowledge required by physicians for accurate prediction of breast cancer recurrence and better decision-making. The main objective of this study is to compare different data mining algorithms to select the most accurate model for predicting breast cancer recurrence. This study is cross-sectional and data gathering of this research performed from June 2018 to June 2019 from the official statistics of Ministry of Health and Medical Education and the Iran Cancer Research Center for patients with breast cancer who had been followed for a minimum of 5 years from February 2014 to April 2019, including 5471 independent records. After initial pre-processing in dataset and variables, seven new and conventional data mining algorithms have been applied that each one represents one kind of data mining approach. Results show that the C5.0 algorithm possibly could be a helpful tool for the prediction of breast cancer recurrence at the stage of distant recurrence and nonrecurrence, especially in the first to third years. also, LN involvement rate, Her2 value, Tumor size, free or closed tumor margin were found to be the most important features in our dataset to predict breast cancer recurrence.

## Introduction

Cancer refers to any one of a large number (over 100 diverse diseases) of diseases characterized by the development of abnormal cells that divide uncontrollably due to genetic mutation in DNA and can infiltrate and destroy normal body tissue. What is similar among all is the deficiency in the regulating mechanisms of the cells' natural growth, proliferation, and death. The cancerous cells can invade the tissues nearby and spread to other parts of the body eventually and is the second-leading cause of death in the world [1, 2].

**Funding:** The authors received no specific funding for this work.

**Competing interests:** The authors have declared that no competing interests exist.

Breast cancer is the most common cancer among women all over the world, and it is increasing 2 percent annually [1, 3–5]. Although the average age of breast cancer is 40–50 years, it has also occurred in 25-year-olds, and the age of onset is declining annually. Concerning the high incidence and prevalence of the disease, and also its high treatment cost, and disease occurrence among young women who are socially and economically generative, this disease can be one of the most remediable diseases if an early diagnosis takes place [6–8]. So, the significance of the prediction and treatment of cancer is more obvious than before [9–11]. The disease occurs when people are exposed to carcinogen materials by inhaling, eating, drinking, and exposing at the workplace and environment. Cancer is a multifactorial disease and beside genetic factors, personal lifestyle, smoking, and diet play impressive effects in the etiology of cancer. There is always a balance among the amount of proliferation, the cell senescence, and the differentiation in a healthy organism, this balance is lost during cancer formation [12–14].

On the other hand, 207 million new cases of breast cancer are likely to be diagnosed in 2030, among which 60% cases are diagnosed in under-developed countries [4, 14, 15]. Moreover, the number of cancer incidence is increasing in these countries.

Breast cancer is one of the most common diseases among Iranian women. Oncological studies show that the prevalence of breast cancer among Iranian women is increasing, and the age of onset of this disease is decreasing with more disease intensity [8, 16]. On the other hand, different factors influence the deterioration of the disease and its overall survival. Because it is complicated to control the causes of breast cancer, early diagnosis along with a suitable treatment is an important strategy for prognosis improvement [11, 17].

In Iran, due to the population growth, the increase of life expectancy, the relative increase of old population, the annual increase of cancer and the acceleration of change among influential factors in the escalation of cancer prevalence such as the prevalence and the development of the cancer risk factors, the rate of cancer development is predicted to rise and it will be doubled during the next two decades [8, 16]. Therefore, this disease and programs to control are of utmost importance. International studies suggest that about 30% of women will develop recurrence after the primary treatment for breast cancer, figures for the early-stage disease being lower [18, 19].

Also, it is the most common malignancy among Iranian women and the focus of attention. In recent years, the prevalence of the disease has been increasing, and data suggest that the survival rate of patients up to five years and ten years after diagnosis was 88% and 80%, respectively. In reality, not all tumors are cancerous, but they may be benign or malignant [6, 13, 20, 21].

Routine follow-up protocols currently are based on two hypotheses: (i) that most recurrences are diagnosed at an earlier stage through follow up; and (ii) that the previous treatment of recurrences offers a better chance of cure, more prolonged survival, or improvement in the quality of life. However, current data suggest that neither of these hypotheses is correct and that postoperative follow up of patients with breast cancer is costly and time-consuming and does not significantly extend survival [18, 22, 23].

Despite considerable effort to detect recurrent disease early, the evidence suggests that only a minority of recurrences are detected at an asymptomatic stage [5, 24–26].

It has also been found that most patients with recurrence have signs or symptoms as the first indicator of recurrence and that history and physical examination generally provide the first clues to recurrence [27, 28].

Furthermore, most recurrences present at unscheduled appointments and not as a consequence of routine follow up ones [29]. Predicting breast cancer recurrence is one of the most popular measures taken for developing data mining approaches.

Physicians make decisions on clinical scenarios (about diagnosis, investigation, and management) every day of their working lives, based on the balance of probabilities [30].

Although randomized trials of intensive surveillance testing such as more frequent clinical examinations, biannual chest x-rays, and bone scans have shown no mortality benefit, there has been a continued rise in financial cost and resource utilization devoted to developing more effective follow-up strategies to detect early recurrences. It's necessary to the application of a multidisciplinary approach to help physicians to make better decisions in the detection of breast cancer recurrent [31].

Data mining methods may help reduce the number of false positives and false-negative results in physicians' decision-making [12, 32]. Accordingly, new approaches, such as knowledge discovery in databases (KDD), including data mining algorithms, have become increasingly popular and a desirable research tool for medical science researchers. Using them, researchers can identify patterns and relationships among a large number of variables, and using the data available in databases has made it feasible to predict the results of a disease [33].

Many studies have been carried out about difficulties in predicting the survival of patients through using statistical methods and artificial neural networks; nevertheless, only a few studies have been conducted in the field of cancer recurrence using data mining methods. Hence, this study compares the prediction of breast cancer recurrence in Iran using data mining methods and suggests a reasonable model for help to predict between several proposed models. According to the importance of the subject, the main purpose of the present study is to assist clinicians and decision-makers in the field of breast cancer for prediction of recurrence using seven data mining algorithms. Therefore, first, the used dataset is introduced. Next, all algorithms and their findings are presented separately. the results of the algorithms are compared to provide recommendations to physicians and decision-makers. And finally, the best features based on data mining algorithms extracted.

## Materials and methods

This study is cross-sectional and data gathering of this research performed from June 2018 to June 2019 from the official statistics of Ministry of Health and Medical Education and the Iran Cancer Research Center for patients with breast cancer who had been followed for a minimum of 5 years from February 2014 to April 2019, including 5471 independent records. for limitation in our dataset (we had only 16 complete information about patients that their relapse occurred after 10 years), we have analyzed only patients with shorter than 5 years.

(early stages of N +, N0, T2, T1) is eligible for inclusion in this study. All patients are confirmed with a pathologic type of ductal carcinoma (Infiltrating ductal carcinoma). Patients presenting with distant metastases (stage IV) and T4 (tumors of any size directly invading the chest wall or skin) at the time of presentation are patients with other histopathologic findings of breast cancer, patients with information in their records. It is incomplete, and Patients who had not undergone lymph node surgery (NX) are excluded.

Also, required data are collected from patients' records, including the status of ER, PR, HER-2 receptors, patient age, tumor size, lymph node involvement status, tumor grade, etc. The condition of the receptors is evaluated by immunohistochemistry.

In the present study, at the first step, the chi-square test is used to evaluate the relation of each of the features on the recurrence of breast cancer, and $P < 0.05$ is considered significant. Also, to develop predictive models and predict breast cancer recurrence, Multilayer Perceptron artificial neural network, Bayesian Neural Network, LVQ neural network, KPCA-SVM, Random Forest, and C5.0 are employed using the above-mentioned database.

## Data mining methods

The process of extraction of the unknown, correct, and potentially useful data is called data mining. The data mining methods can be considered as unsupervised and supervised learning. In this study, seven different types of neural networks are exploited for the recurrence of breast cancer prediction. the number of inputs to the network is 17 key variables extracted from 23 variables, which is shown in Table 1. Fig 1 represents the algorithm used in the current study. According to the depicted algorithm, first, the data are grouped into test and train data; the results are compared and finally, the best features are selected.

The accuracy of the model is the percentage of the number of times the test samples are successfully categorized. If the model accuracy is acceptable, the model can be used to classify data whose categories are not specified. In this research, we use a nested 5-fold cross-validation approach to train (four folds) and test (one fold) the models. Patients meeting the inclusion criteria are randomly assigned to one of the five outer folds. To ensure that the important feature set is generated from real patients with breast cancer and the importance of the features is not emphasized by duplicating minor cases, we choose the under-sampling approach to build the model. We randomly select 20 sets of controls in each round of cross-validation, matching the number of cases, and generated 80 training datasets by using one set of controls and all cases. In each training step, we use 5-fold inner cross-validation to tune the models. this method is used but for preventing of exceeding machine learning analysis. Also, to assess the performance of disease classification methods, a confusion matrix used. This matrix is for the accuracy of the obtained model is calculated. The following formula applies to calculate efficiency, sensitivity, and specificity. The results show the numerical values obtained from the calculations performed by the software and its final analysis. Models performance is evaluated according to the following criteria:

**Feature.** Probability of correctly predicting non-recurrence by true negative algorithms divided by false-positive + true negative.

**TP.** The number of samples that are correctly identified as positive

**TN.** The number of samples that are correctly diagnosed as negative.

**FP.** The number of samples that incorrectly detected positive

**FN.** The number of samples that incorrectly detected negative

**Confusion matrix.** The relationship between the actual classes and the predicted classes is called the confusion matrix.

**Accuracy.**   This is the number of samples that are correctly identified, relative to the total sample.

$$\text{Accuracy} = (TP + TN)/(TP + TN + FP + FN)$$

**Sensitivity.**   The probability of correctly predicting recurrence by true positive algorithms divided by false-negative + true positive.

$$\text{Sensitivity} = TP/(TP + FN)$$

**Specificity.** Measures a test's ability to correctly generate a negative result for people who don't have the condition that's being tested for (also known as the "true negative" rate).

$$\text{Specificity} = TN/(TN + FP)$$

**Table 1. Variables relating to the recurrence of breast cancer in a dataset.**

| No | Factor | Scale | Study | p-value |
|---|---|---|---|---|
| 1 | Diagnosis of age | 0 = "<35" | [36–38] | 0.207 |
| | | 1 = "35–44" | | |
| | | 2 = "45–55" | | |
| | | 3 = ">55" | | |
| 2 | Menarche age | 0- after the age of 12 | [36, 39] | 0.621 |
| | | 1- age of 12 ≥ (risky) | | |
| 3 | Menopause age | 0- before the age of 50 | [36, 39–41] | 0.934 |
| | | 1- ≥ age of 50 (risky) | | |
| | | 2- menopause has not yet occurred | | |
| 4 | History of infertility | 0- No | [42, 43] | 0.201 |
| | | 1- Yes | | |
| 5 | Family history of breast cancer | 0- No | [35] | 0.325 |
| | | 1- Yes | | |
| 6 | Family history of other cancers | 0 = "no"  3 = "colon cancer" | [44] | 0.146 |
| | | 1 = "mail breast cancer"  4 = "ovarian cancer" | | |
| | | 2 = "prostate cancer"  5 = "uterus cancer" | | |
| 7 | Tumor site | 1 = "uoq" 12 = "upper half" | [34, 40, 45, 46] | 0.230 |
| | | 2 = "uiq" 13 = "latral half" | | |
| | | 3 = "loq" 14 = "uoq and liq" | | |
| | | 4 = "liq" 24 = "medial half" | | |
| | | 5 = "central(nipple areole)" 30 = "three quadrant" | | |
| | | 6 = "axilla" 34 = "lower half" | | |
| | | 50 = "diffuse" | | |
| 8 | Tumor side | 1- right breast tumor | [45–47] | 0.514 |
| | | 2- left breast tumor | | |
| | | 3- bilateral tumor | | |
| 9 | Tumor size | 1 = "<2" | [45–47] | 0.002 |
| | | 2 = "2.5" | | |
| | | 3 = ">5" | | |
| | | 4 = "chest wall or skin" | | |
| 10 | LN involvement rate | 0 = "no" | [34, 47, 48] | 0.001 |
| | | 1 = "1–3" | | |
| | | 2 = "4–9" | | |
| | | 3 = "0>9" | | |
| 12 | Tumor site | 1 = "local" 12 = "local & axilla" | [45–47, 49] | 0.270 |
| | | 2 = "axilla" 13 = "local & regional" | | |
| | | 3 = "regional" 14 = "local & local (post MRM)" | | |
| | | 4 = "Local (post MRM)" | | |
| 13 | Result of biopsy of pathology | 1 = "lcis" 7 = "paget's disease" | [38, 50, 51] | 0.018 |
| | | 2 = "dcis" 8 = "others" | | |
| | | 3 = "ilc" 9 = "inflammatory carcinoma" | | |
| | | 4 = "ilc" 11 = "sarcoma" | | |
| | | 5 = "medullary" 12 = "metastatic(un known origin)" | | |
| | | 6 = "microinvasion" 13 = "lymphoma" | | |
| 14 | Type of surgery | 1 = "MRM" | [16, 23, 52] | 0.001 |
| | | 2 = "Breast preservation" | | |
| | | 3 = "bilatral MRM" | | |
| | | 4 = "bilatral BCS" | | |
| | | 5 = "bilatral MRM & BCS" | | |

*(Continued)*

**Table 1.** (Continued)

| No | Factor | Scale | Study | p-value |
|----|--------|-------|-------|---------|
| 15 | Tumor grade | 1 = "1" | [19, 45–47] | 0.093 |
|    |             | 2 = "2" | | |
|    |             | 3 = "3" | | |
| 16 | Free or closed tumor margin | 0 = "free (> = 2cm)" | [45–47] | 0.001 |
|    |             | 1 = "closed (< = 2cm)" | | |
|    |             | 2 = "involve" | | |
| 17 | Estrogen Receptor value | 0- Negative | [53] | 0.405 |
|    |             | 1- Positive | | |
|    |             | Positive value increases recurrence risk | | |
| 18 | Progesterone Receptor value | 0- Negative | [10, 28, 54] | 0.181 |
|    |             | 1- Positive | | |
|    |             | Positive value increases recurrence risk | | |
| 19 | Her2 value | 0- Negative | [55] | 0.312 |
|    |             | 1- Positive | | |
|    |             | A positive value increases recurrence risk | | |
| 20 | Chemotherapy | 0 = "no" | [56] | 0.002 |
|    |             | 1 = "yes" | | |
|    |             | 19 = "chemo + Herceptin" | | |
| 21 | Type of chemotherapy (Typechemo) | 1 = "neo adjvant" | [50, 53, 56] | 0.161 |
|    |             | 2 = "adjvant" | | |
| 22 | Radiotherapy | 0- No radiotherapy | [57, 58] | 0.000 |
|    |             | 1- Radiotherapy has been performed | | |
|    |             | When no radiotherapy has been performed, cancer recurrence is more likely to occur. | | |
| 23 | Hormone therapy | 0 = "no" | [58, 59] | 0.001 |
|    |             | 1 = "tamoxifen" 4 = "aromazin (exemstane)" | | |
|    |             | 2 = "raloxifen" 5 = "megace" | | |
|    |             | 3 = "femara or letrozol" 6 = "others" | | |

**ROC.** The method based on the receiver operating characteristic curve [9, 34, 35] is used to evaluate the performance of different discriminant models. The area under the ROC curve (AUC) can objectively reflect the overall performance of different algorithms.

**F-score.** In the statistical analysis of binary classification, the **$F_1$ score** (also **F-score** or **F-measure**) is a measure of a test's accuracy. It considers both the precision) p and the recall) r of the test to compute the score: p is the number of correct positive results divided by the number of all positive results returned by the classifier. R is the number of accurate positive results divided by the number of all relevant samples (all samples that should have been specified as positive). The $F_1$ score is the harmonic mean of the precision and recall, where an $F_1$ reaches its best value at 1 (perfect precision and recall) and its worst at 0.

## Ethics statement

All data were fully anonymized before we accessed them and the Ministry of Health and Medical Education and the Iran Cancer Research Center are informed consent. All the private information of patients is eliminated from our dataset and only clinical and required data is used and presented.

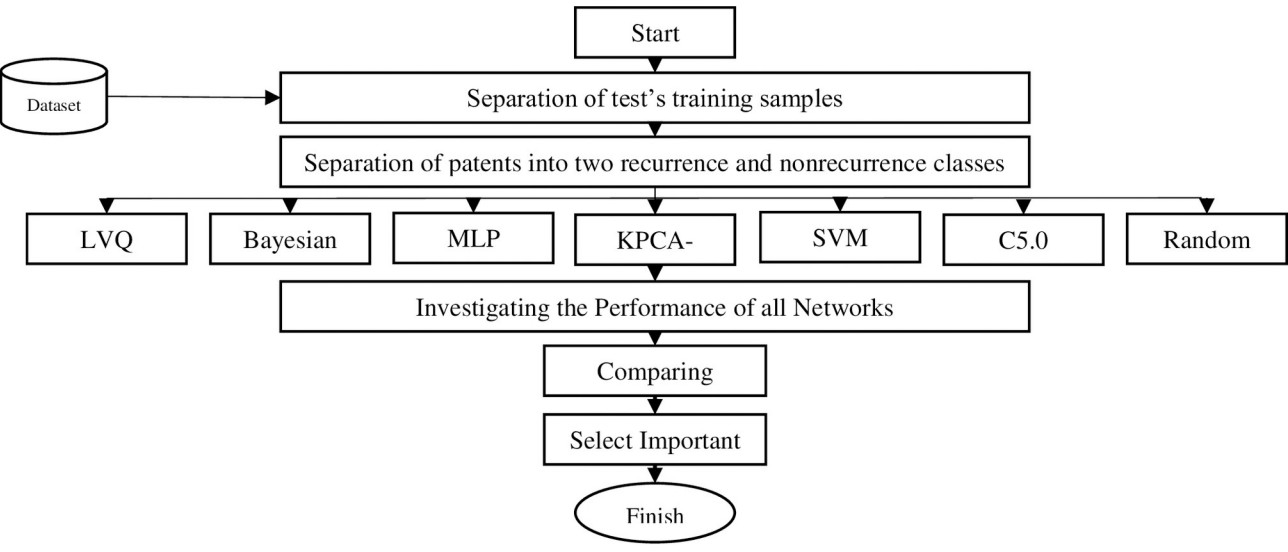

**Fig 1. The exploited algorithm in this study.**

## Results

In the present study, seven data mining algorithms based on the accuracy, Sensitivity, Specificity, F-measure, area under the ROC curve of the models, etc. are compared. The final goal is to achieve a model with the highest performance. So, insufficient and defective data is identified and eliminated from the database. Concerning the elimination of variables, variables either overlapped with the results or that among 10337 records available, their information on these variables was eliminated. Working on the documents is begun after deleting the variables according to the abovementioned two modes (overlapping with other variables or lacking maximum record information from this variable). For patients with no data or data limited to 2–3 cases, we eliminate the record inevitably. The remaining documents with fewer missing variables are replaced by expectation-maximization.

After the elimination of records with missing values, 5471 records remained. For the best prediction, data mining classification techniques including Multilayer Perceptron artificial neural network, Bayesian Neural Network, LVQ neural network, KPCA-SVM, C5.0, and Random Forest are applied. Additionally, several parameters are randomly selected and investigated.

In this study, breast cancer is classified into four groups based on IHC profile ER/PR and Her2/neu expression, positive (+), and/or negative (−). The groups are:

ER/PR+, Her2+ = ER+/PR+, Her2+; ER−/PR+, Her2+; ER+/PR−, Her2+

ER/PR+, Her2− = ER+/PR+, Her2−; ER−/PR+, Her2−; ER+/PR−, Her2−

ER/PR−, Her2+ = ER−/PR−, Her2+

ER/PR−, Her2− = ER−/PR−, Her2−

The IHC classification correlates well with intrinsic gene expression microarray categorization: ER/PR+, Her2+ with Luminal B; ER/PR+, Her2− with Luminal A; ER/PR−, Her2+, and ER/PR−, Her2− with triple-negative/basal-like tumors. Apart from lending itself to subtype analyses of tumors when fresh tissue is not available, the IHC classification has prognostic and therapeutic implications, is inexpensive and readily available.

According to [11, 26] studies, and available data of Ministry of Health and Medical Education and the Iran Cancer Research Center's, patients classified in four segments: Local

**Table 2.**

| Demographic and clinical characteristics | | Percent | Frequency |
|---|---|---|---|
| Age (years) | <50 | 67.4 | 3687 |
| | >50 | 32.6 | 1784 |
| Tumor size | T1* | 26.8 | 1466 |
| | T2* | 55.1 | 3015 |
| | T3* | 18.1 | 990 |
| Tumor Grade | 1 | 16.7 | 914 |
| | 2 | 58 | 3173 |
| | 3 | 25.4 | 1390 |
| Lymph node metastasis (number) | Negative | 36.2 | 1981 |
| | Positive<3 | 31.1 | 1701 |
| | Positive >3 | 32.7 | 1789 |
| Estrogen receptor | Negative | 31.2 | 1707 |
| | Positive | 68.8 | 3764 |
| Progesterone receptor | Negative | 36.2 | 1981 |
| | Positive | 63.8 | 3490 |
| HER-2 receptor | Negative | 47.1 | 2577 |
| | Positive | 52.9 | 2894 |
| Triple-negative | | 13.7 | 750 |

T1 *: A tumor with a diameter of 2 cm or less

T2 *: Tumor diameter between 2 and 5 cm

T3 *: A tumor the largest diameter of which is over 5 cm

Recurrence, Regional recurrence, Distant recurrence, Local and Distant recurrence, Reginal and Distant recurrence and No recurrence. The number of patients in each category is 607, 82, 1975, 339, and 65.

The demographic and clinical characteristics of the units under study are shown in Table 2 and Table 3. The mean and standard deviation of the age is 46.86 ± 10.397 years, and the age group, 41–50 years with 42.8%, had the highest frequency. The mean tumor diameter is 3.63 ± 1.86 cm. The rate of demographic and clinical factors in the studied patients is depicted in Table 2.

Some characteristic of patients are presented in Table 4 and Table 5. The median local recurrence time of patients is 16 months and the median metastasis time is 22 months. The most common sites of bone metastases (46.6%) are with a preference for the spine, especially the lumbar vertebrae. The median survival time from illness was 36 months. The overall treatment rate of the disease in the fifth year is 45%. This rate is 85% in the first stage of the disease and 5% in the fourth stage.

Variables relating to the recurrence of breast cancer based on various studies as well as interviews with specialists in the field of breast cancer are obtained as Table 1. also, the relationship of each factor with the recurrence of breast cancer in our dataset is evaluated by using Chi-Squared test. factors with a p-value of less than 0.05 are considered as the most relating factors.

Based on this Table, the most relating factors include Tumor size, LN involvement rate, the result of biopsy of pathology, type of surgery, free or closed tumor margin, Chemotherapy, Radiotherapy, and Hormone therapy.

By refining the results and discarding the treatment and diagnosis methods, 3 factors of Tumor size, LN involvement rate, and free or closed tumor margin are considered as the most important factors affecting the recurrence of breast cancer in our dataset.

**Table 3. Frequency distribution of demographic and clinical characteristics of patients.**

| Patient Description | Receptors status | Non-TN | TN | HER- | HER+ | PR- | PR+ | ER- | ER+ |
|---|---|---|---|---|---|---|---|---|---|
| | | Number(Percent) | | Number(Percent) | | Number(Percent) | | Number(Percent) | |
| Year | <40 | 481(8.8) | 317(5.8) | 673(12.3) | 673(12.3) | 635(11.6) | 164(3) | 635(11.6) | 711(13) |
| | 41–50 | 2183(39.9) | 159(2.9) | 832(15.2) | 1505(27.5) | 553(10.1) | 1784(32.6) | 514(9.4) | 1822(33.3) |
| | >51 | 1505(27.5) | 2937(5.1) | 1078(19.7) | 711(13) | 793(14.5) | 996(18.2) | 558(10.2) | 1231(22.5) |
| Tumor Size | T1* | 1269(23.2) | 197(3.6) | 673(12.3) | 793(14.5) | 514(9.4) | 952(17.4) | 399(7.3) | 1072(19.6) |
| | T2* | 2697(49.3) | 317(5.8) | 1308(23.9) | 1707(31.2) | 990(18.1) | 2024(37) | 914(16.7) | 2095(38.3) |
| | T3* | 755(13.8) | 235(4.3) | 596(10.9) | 394(7.2) | 476(8.7) | 514(9.4) | 394(7.2) | 597(10.9) |
| Tumor Grade | 1 | 793(14.5) | 120(2.2) | 558(10.2) | 356(6.5) | 159(2.9) | 755(13.8) | 159(2.9) | 755(13.8) |
| | 2 | 2736(50) | 438(8) | 1625(29.7) | 1548(28.3) | 1149(21) | 2025(37) | 908(16.6) | |
| | 3 | 1187(21.7) | 197(3.6) | 394(7.2) | 990(18.1) | 673(12.3) | 711(13) | 635(11.6) | 755(13.8) |
| Lymph node metastasis | Positive | 3091(56.5) | 399(7.3) | 1784(32.6) | 1707(31.2) | 1149(21) | 2342(42.8) | 2342(18.8) | 2456(44.9) |
| | Negative | 1624(29.7) | 356(6.5) | 793(14.5) | 1187(21.7) | 832(15.2) | 1149(21) | 678(12.4) | 1308(23.9) |

T1 *: A tumor with a diameter of 2 cm or less

T2 *: Tumor diameter between 2 and 5 cm

T3 *: A tumor the largest diameter of which is over 5 cm

Then, to the prediction of breast cancer recurrence, by discarding 6 factors and of Chemotherapy, Type of chemotherapy, Radiotherapy, Hormone therapy, and Result of biopsy of pathology, other features are considered as the inputs of machine learning algorithms.

## Modeling using multilayer perceptron neural network

In the first step, a multilayer perceptron (MLP) neural network is used, which is one of the purest and most potent structures for modeling.

The output layer function is defined by (1), in which $o$ and $h$ represent the hidden layer and output layer, respectively, and $w$ is the weights of the layers.

$$O_i = sgm\left(\sum_m sgm\left(\sum_i x_i w_{lm}^h\right) w_{mi}^o\right) \tag{1}$$

In this equation, *sgm* is a sigmoid function defined as:

$$sgm(x) = \frac{1}{1 + e^{-x}} \tag{2}$$

The first modeling method is a multilayer perceptron neural network with backpropagation error algorithm for training, which is used for classification (recurrence and non-recurrence of breast cancer), and all of the eight risk factors are fed to the network. Also, different network architectures are evaluated to acquire the best classification performance. Table 6 shows some of the best structures of the multilayer perceptron neural network, and the confusion matrix related to the best architecture (among almost 100 different architectures).

**Table 4. Confusion matrix.**

| Predicted Values | Actual Values | | |
|---|---|---|---|
| | | positives | negative |
| | positives | TP | FP |
| | negative | FN | TN |

**Table 5. Frequency distribution of ER and PR receptors in terms of HER-2 receptors.**

| HER2 receptor status / ER and PR receptor status | HER- Number(Percent) | HER+ Number(Percent) |
|---|---|---|
| ER+ | 1822(33.3) | 1942(35.5) |
| ER- | 755(13.8) | 952(17.4) |
| PR+ | 1625(29.7) | 1866(34.1) |
| PR- | 952(17.4) | 1029(18.8) |

## Modeling using learning vector quantization

Learning Vector Quantization (LVQ) neural network consists of a competitive layer and a linear layer. The competitive layer learns to classify the input vectors, and the linear layer maps the competitive classes on the target category, which is determined by the user.

In this study, 10 LVQ neural networks, with different architectures, are used to find the optimum network architecture for disease Recurrence. Table 7 shows the performance of the various neural networks over the changes in the number of neurons in the competitive layer.

## Modeling using Bayesian network

The Bayesian network is a decision-making system and has a strong ability to model the cause and effect relationship. Note that it is not necessary a Bayesian neural network that has precise information and a complete history of an event, and it can reach a convincing estimate of the current or future condition of a system even with incomplete and not precise information. Therefore, in this paper, the Bayesian neural network is exploited as the third method for disease classification. The distribution of the weights is defined by (3).

$$y_k = f_{outer}\left(\sum_{j=1}^{m} w_{kj}^{(2)} f_{inner}\left(\sum_{i=1}^{d} w_{ji}^{(1)} + w_{j0}^{(1)}\right) + w_{k0}^{(1)}\right) \tag{3}$$

$$P(W|D) = \frac{P(D|W)P(w)}{P(D)} \tag{4}$$

Where $w_{ji}^{(1)}$ and $w_{ki}^{(2)}$ are the weights of the first and second layers, respectively, $i$, $j$, and $k$ are the indexes of input, hidden layer, and output, respectively, and $w_{j0}^{(1)}$ is the bias for the $j^{th}$ hidden unit. M is the number of hidden units; $d$ is the number of input units. The function $f_{outer}(.)$ is linear, and $f_{inner}(.)$ is a hyperbolic tangent function. In the Bayesian method, Eq (3) used weight distribution, and the network weights are computed by (5).

$$P(W|D) = \frac{P(D|W)P(w)}{P(D)} \tag{5}$$

*Where p(w) is the probability distribution function in the weight space without data, which is a primary function. P(W|D) is the probability function of weights that is a probability*

**Table 6. The test of different architectures of MLP neural network.**

| Number of hidden layers | Neural network architecture |
|---|---|
| 2 | 1-10-10 |
| 2 | 1-10-20 |
| 3 | 1-10-10-10 |

**Table 7. The result of the LVQ algorithm for the number of neurons.**

| The number of neurons | 3 | 4 | 5 | 6 | 7 |
|---|---|---|---|---|---|
| Number of loop repeats | 4 | 4 | 4 | 2 | 2 |

distribution function observed after the train of the data. *P(D|W)* and *P(D)* is the probability distribution function and the second probability distribution function, respectively.

Different structures are tested to find the best number of neurons in the competitive layer. Some of them are shown in Table 8. According to the table, the first four architectures have two hidden layers. Increasing the number of hidden layers from two to three (the last two rows of the table) does not have a significant effect on error performance (only 0.3% improvement).

## Modeling using KPCA- SVM

To further improve the discrimination accuracy, combining kernel principal component analysis and support vector machine (KPCA-SVM) is proposed in this study. After preprocessing, the dataset is analyzed with the model of KPCA-SVM. The results show that the model can achieve excellent results.

For KPCA, by introducing the kernel function, the original dataset can be mapped into a high-dimensional space [60–62]. Record the original spatial dimension as $R^n$, the number of samples is $m$, and the dimension is $n$, then the total sample is $X = [x_1, x_2, ..., x_m]$, $x_i = [x_{i1}, x_{i2}, ..., x_{im}]$, $i = 1, 2, ..., m$. Defining has a high-dimensional Hilbert space, all the samples are mapped from the original space to the high-dimensional space, which is expressed as:

$$x_i \rightarrow \Phi(x_i) \tag{6}$$

The kernel function is defined as

$$K = \Phi(X)^T \Phi(X) = [k(x_i, x_j)]_{m \times m,} \tag{7}$$

Which needs to satisfy the Mercer condition

$$k(x_i, x_j) = \langle \Phi(x_i)^T, \Phi(x_j) \rangle = \Phi(x_i)^T, \Phi(x_j). \tag{8}$$

After nonlinear mapping, the corresponding covariance matrix is

$$the\ matrix\ is\ C = \frac{1}{m} \sum_{i=1}^{m} \Phi(x_i)^T, \Phi(x_i). \tag{9}$$

Denote the eigenvalues and eigenvectors as λ and V, respectively

$$\lambda V = CV, \tag{10}$$

Where V is the subspace generated by $\{\Phi(x_1), \Phi(x_i), ..., \Phi(x_m)\}$.

**Table 8. A survey of different neural network architectures considering all input parameters in the recurrence of breast cancer disease.**

| Neural network architecture | Number of hidden layers |
|---|---|
| 4-6-1 | 2 |
| 8-7-1 | 2 |
| 13-6-1 | 2 |
| 13-18-1 | 2 |
| 12-6-4-1 | 3 |
| 18-8-6-1 | 3 |

Then there exists $\alpha = \{\alpha_1\alpha_2\ldots\alpha_m\}$ making

$$V = \sum_{j=1}^{m} \alpha_j \Phi(x_j). \tag{11}$$

The following transformation is made for Eq (10), when $K = 1,2,\ldots,m$

$$\lambda(\Phi(x_k)V) = \Phi(x_k)CV. \tag{12}$$

Mixing Eqs (9), (11) and (12), we get

$$\lambda(\Phi(x_k)\sum_{j=1}^{m} \alpha_j\Phi(x_j)) = \Phi(x_k)\frac{1}{m}\sum_{i=1}^{m}\Phi(x_i)^T\Phi(x_i)\sum_{j=1}^{m}\alpha_i\Phi(x_j). \tag{13}$$

Further, we get

$$\lambda\sum_{j=1}^{m}\alpha_j(\Phi(x_k)\Phi(x_j)) = \frac{1}{m}\sum_{j=1}^{m}\alpha_j(\Phi(x_k)\sum_{i=1}^{m}\Phi(x_i))(\Phi(x_i)^T\Phi(x_j)). \tag{14}$$

When we substitute Eqs (8) and (9) into Eq (15), yield

$$m\lambda\alpha = K\alpha \tag{15}$$

Thereby the eigenvalues and eigenvectors of the K matrix can be obtained.

## Modeling using support vector machines

Support Vector Machines (SVM) has unique advantages and has been widely used in the field of pattern recognition, especially in dealing with classification problems.

For the two-category problem, set the given training sample set as $\{(x_1,y_1),(x_2,y_2)\ldots,(x_n,y_n)\}$,

Which $y_i\epsilon\{+1, -1\}(i = 1,2,\ldots,n)$ is the category the sample belongs to Construct a cost function that satisfies the constraint as

$$\min\frac{1}{2}\|\omega\|^2 + C\sum_{i=1}^{n}\xi_i, \tag{16}$$

$$s.t. y_i(\omega^T x_i + b) \geq 1 - \xi_i, \xi_i \geq 0, i = 1, 2, \ldots, n.$$

Where $\xi$ is the introduced slack variable to measure the misclassification degree of the model, C is the penalty constant, which should be selected appropriately, $\omega$ and $b$ are the weight vector and threshold of the classification function, respectively. Lagrangian function is

$$L(\omega, b, \alpha) = \frac{1}{2}\|\omega\|^2 + C\sum_{i=1}^{n}\xi_i - \sum_{i=1}^{n}\beta_i\xi_i - \sum_{i=1}^{n}\alpha_i[y_i(\omega^T.x_i + b) - 1 + \xi_i] \tag{17}$$

Where $\alpha_i$ and $\beta_i$ are Lagrangian operators. According to Karush-Kuhn-Tucher conditions, we have

$$0 \leq \alpha_i \leq C, i = 1, 2, ..,n, \tag{18}$$

$$\sum_{i=1}^{n}\alpha_i y_i = 0, \tag{19}$$

Where $\alpha_i > 0$ is a support vector. The discriminant function is

$$f(x) = sgn(\sum_{i=1}^{n} \alpha_i^* y_i K(x, x_i) + b^*) \tag{20}$$

where K is the selected kernel function.

Fig 2 shows the workflow of the Raman spectral discrimination model of KPCA-SVM.

## Modeling using random forest

Random forest is a combination of learning trees that each tree in the forest is built from a random vector called Q. The $Q_k$ vector describes the method of constructing the kth tree. For

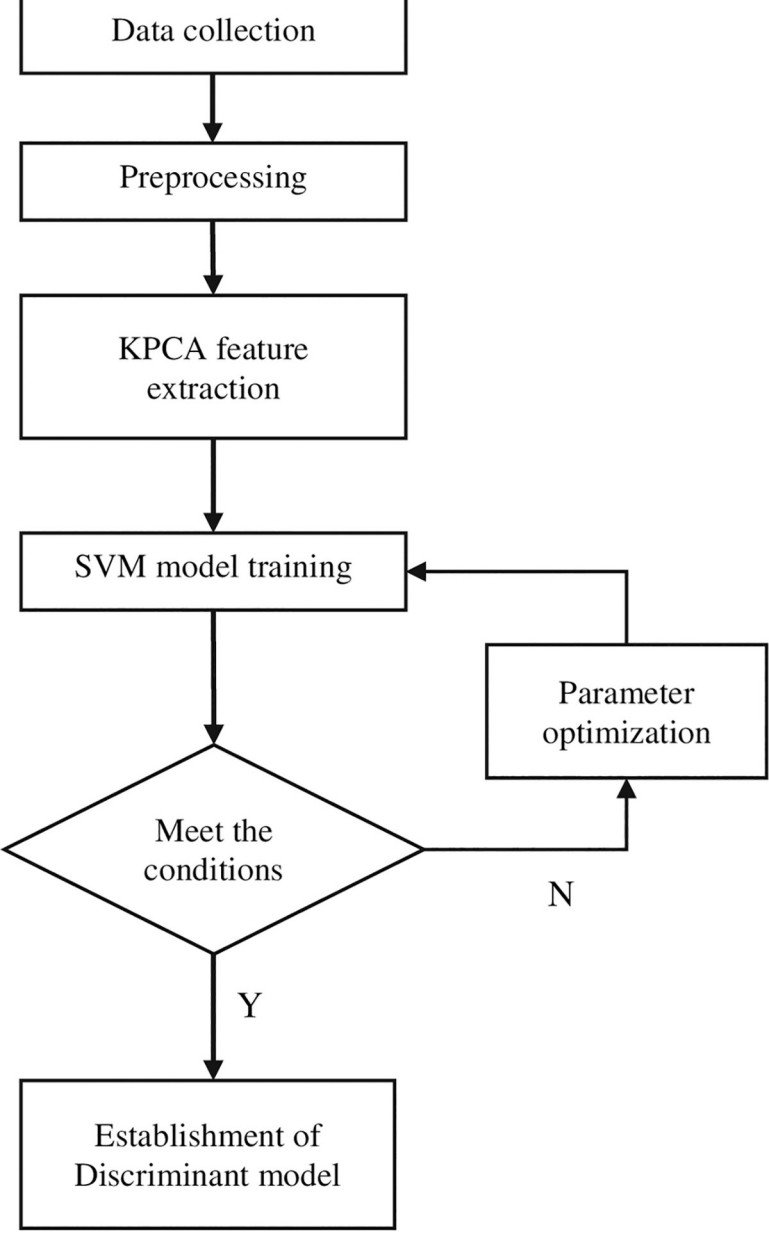

**Fig 2. The workflow of the Raman spectral discrimination model of KPCA-SVM.**

example, it may specify whether some features are randomly used in the construction of each tree, or whether the training data set is randomly selected. Each tree constructed with h (x, $Q_k$) is represented. This vector is given to all trees to classify the input x,h (x, $Q_k$), k = 1,..., k, and is the final class of which with the most trees to vote on.

Suppose $h_1$ (x),..., $h_k$ (x) are some classifiers. Suppose Y is the output vector of the training data, and X is a random vector drawn from the training data. Now the boundary function of this set of classifiers is defined as follows:

$$mg\left(\underline{X}, Y\right) = \frac{\sum_{k=1}^{k} I(h_k(\underline{X}) = Y)}{k} - \max_{j \neq Y}\left[\frac{\sum_{k=1}^{k} I(h_k(\underline{X}) = j)}{k}\right] \qquad (21)$$

Formula 22 Calculates the Class Boundary Function

Where I (-) is the marker function. This relation shows that if the data X is given as an input to the k cluster, it is the number of classifications performed by the k cluster correctly, more or less than the other classifications performed.

If mg (X, Y)> 0, the set of classifiers will be correctly sorted. If mg (X X, Y) <0, then the classification is incorrect.

The generalization error is defined as follows:

$$PE^+ = P_{\underline{X}}, Y(mg(\underline{X}, Y) < 0) \qquad (22)$$

In the random forest i.e. h (x x, ((Q_k)) k = 1,..., k it is shown in [57] study that when a random forest grows and k tends to infinity:

$$PE^+ \rightarrow P_{\underline{X}}, Y(P_Q(h(\underline{X}, Q) = Y) - \max_{j \neq Y} P_Q(h(\underline{X}, Q) = j) < 0), k \rightarrow \infty \qquad (23)$$

Formula 24 Classification General Error When k→∞

It indicates that the generalization error tends to be above the limit. Therefore, it does not over-fit the data.

## Modeling using decision tree C5.0

A Decision tree is a powerful method among classification algorithms whose popularity is increasing with the growth of data mining. Decision trees classify samples by sorting them in the tree from the root node to the leaf node. The specimens are grouped in such a way that they grow from the root downwards and eventually reach the leaf nodes. Each inner or non-leaf node is specified with a property. This feature raises a question about the input example. There are many possible answers to this question in each internal node as it's specified by its value. The leaves of this tree are characterized by a class or a bunch of answers. The reason for naming it with the decision tree is that it represents the decision-making process for classifying an input instance. The decision tree is used to solve problems that can be presented in a single answer in the name of a class or category. It is also appropriate for problems where the training examples are specified in pairs (value-specificity). The target function has output with discrete values. For example, each sample should be marked with a yes or no or require a seasonal descriptor. The decision tree consists of several algorithms such as C5.0, C4.5, ID3, and classification that used the C5.0 method in this research. The decision tree is used to approximate discrete functions.

It is resistant to input noise. It is useful for high volume data hence in data mining. The tree can be represented as an if-then rule that is understandable for use. It allows the combination of (AND) and seasonal (OR) hypotheses. It is also applicable when tutorials lack all the features.

## Feature importance analysis

We evaluate the importance of features by two methods. First, as shown in Table 1, variables relating the recurrence of breast cancer extracted with chi-square and after refining the results and discarding the treatment and diagnosis methods, 3 factors of Tumor size, LN involvement rate and free or closed tumor margin were considered as the most important factors affecting the recurrence of breast cancer in our dataset.

Second, in data mining algorithms we apply forward feature selection method to validate most influencing factors that affect the recurrence of breast cancer. Forward selection is an iterative method in which we start with having no feature in the model. In each iteration, we keep adding the feature which best improves our model till an addition of a new variable does not improve the performance of the model.

## Performance metrics

The performance of the Seven Methods is evaluated by accuracy, sensitivity, specificity, AUC, PPV, NPV (Table 9). Also, a graphical comparison of the efficiency measures is shown in Fig 3.

As seen in Table 9 and Fig 3, almost all the methods except MLP, Bayesian, and LVQ generate relatively high Sensitivity (more than 70%). Also, the highest Accuracy in our experiments is obtained with C5.0 and KPCA-SVM. Thus, the C5.0 and KPCA-SVM approach appears to perform better than the other five data mining methods. The results indicate that all of the seven methods exceptions for MLP, Bayesian, and LVQ can predict the recurrence of breast cancer, and it is realized that despite adding the new methods to the model, the C5.0 and KPCA-SVM is the best approximation method.

According to the confusion matrix, the final number of patients with recurrence of breast cancer, as well as the patients with no recurrence of the disease, in comparison with the predictions made by different algorithms, is presented in Table 10.

Because the occurrence of errors in classes and forecasting models is inevitable, so the errors in the system should be recognized and investigated. In this study, the confusion matrix has been used to address this important issue. This matrix was generated for all classification models. However, only four confusion matrix for high-precision classifiers are shown below:

Based on the Table 11, the prediction accuracy in each of the following is as follows: The results of Table 12 show that the predictive accuracy of distant recurrence (0.87) and

**Table 9. Performance measure.**

| Method | Random Forest | LVQ | Bayesian | C5.0 | MLP | KPCA-SVM | SVM |
|---|---|---|---|---|---|---|---|
| TP | 1750 | 1640 | 1650 | 2188 | 1477 | 2048 | 1750 |
| TN | 2188 | 2024 | 2008 | 2297 | 1914 | 2250 | 2243 |
| FP | 985 | 931 | 758 | 657 | 657 | 765 | 985 |
| FN | 548 | 876 | 1208 | 329 | 1423 | 408 | 493 |
| Accuracy | 0.719 | 0.669 | 0.650 | 0.819 | 0.619 | 0.785 | 0.729 |
| Sensitivity | 0.761 | 0.651 | 0.577 | 0.869 | 0.509 | 0.833 | 0.780 |
| Specificity | 0.689 | 0.684 | 0.725 | 0.777 | 0.744 | 0.746 | 0.694 |
| The Geometric mean of sensitivity and specificity | 0.724 | 0.668 | 0.647 | 0.822 | 0.615 | 0.788 | 0.736 |
| PPV | 0.639 | 0.637 | 0.685 | 0.769 | 0.692 | 0.728 | 0.639 |
| NPV | 0.799 | 0.697 | 0.624 | 0.874 | 0.573 | 0.846 | 0.819 |
| The Geometric mean of PPV and NPV | 0.715 | 0.667 | 0.654 | 0.820 | 0.630 | 0.785 | 0.724 |
| F-measure | 0.695 | 0.644 | 0.626 | 0.816 | 0.586 | 0.777 | 0.703 |
| The area under ROC curve | 0.729 | 0.632 | 0.692 | 0.763 | 0.625 | 0.774 | 0.742 |

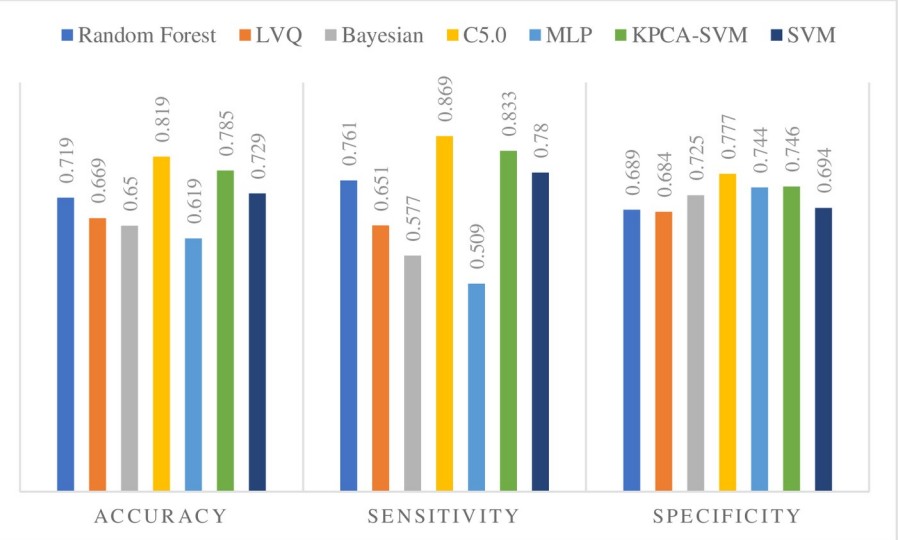

**Fig 3. Performance criteria of the seven classification methods.**

nonrecurrent (0.88) are the most. Also, considering that distant recurrence is in step 7 of the TNM rating, it can be concluded that in this stage, the present research model provides better help to prediction rather than other data mining algorithms.

There is also a general increase in the probability of disease-free survival in the first to fifth years after initial treatment in patients with breast cancer.

According to the results of Table 13, in the first to third years, the further we go in the third year, the more accurate the probability of the disease-free survival becomes, but after the third year, this rate decreases. Therefore, it can be stated that the model presented in this study has higher accuracy in the first three years.

### Important features for breast cancer metastasis prediction

Table 14 represents the results of applying the Forward Selection algorithm on the Ministry of Health and Medical Education and the Iran Cancer Research Center dataset. The most important features affecting the diagnosis of breast cancer recurrence were then identified.

Since the LN involvement rate is the most accurate one in the recurrence of breast cancer detection, the rest of the features are eliminated, the LN involvement rate is selected, and the remaining features are added to it, and the accuracy of the models is measured. In the next step, the set of two features with the highest accuracy are selected and the remaining features are added to them and the accuracy of the model is measured. Then the set of three features with the highest accuracy are selected and the remaining features are added to them and the accuracy of the model is measured.

In the next step, the set of four features with the highest accuracy are selected and the remaining features are added to them and the accuracy of the model is measured. As observed

**Table 10. Actual and predicted recurrence in the studied patients.**

| Actual Positive Recurrence(n) | Actual Negative Recurrence(n) | Predicted Positive Recurrence(n) | Predicted Negative Recurrence(n) |
|---|---|---|---|
| 2517 | 2954 | 2845 | 2626 |

**Table 11. Cancer recurrence matrix of the C5.0 algorithm.**

| predicted Actual | Local recurrence | Regional recurrence | Distant recurrence | Local and Distant recurrence | Regional and Distant recurrence | No recurrence |
|---|---|---|---|---|---|---|
| Local recurrence | 463 | 37 | 6 | 12 | 21 | 68 |
| Regional recurrence | 4 | 54 | 3 | 3 | 8 | 10 |
| Distant recurrence | 3 | 22 | 1724 | 16 | 23 | 187 |
| Local and Distant recurrence | 2 | 18 | 14 | 234 | 19 | 52 |
| Regional and Distant recurrence | 1 | 4 | 2 | 2 | 46 | 10 |
| No recurrence | 64 | 41 | 164 | 40 | 19 | 2626 |

in Table 14, since the accuracy has not increased in all models by adding new features to the set of four features obtained in the previous step, the tests are not continued and it is recognized that in the C5.0 with four properties of {LN involvement rate, Her2 value, Tumor size, free or closed tumor margin} can better help rather than other data mining algorithms to the prediction of breast cancer recurrence on the Ministry of Health and Medical Education and the Iran Cancer Research Center dataset may be made.

As observed in the results of Table 14, in addition to identifying the most important features, the same or better performance was achieved with fewer features than all other features. In machine learning methods, achieving similar or better results using fewer features is important.

The results are consistent with that showed in Table 14, about three factors, and the machine learning method considers a Her2 value factor as an important factor, In addition to those factors, for better accuracy. Therefore, it is recommended that all four factors be considered for better prediction of breast cancer recurrence.

## Conclusion and discussion

Numerous tests for breast cancer often increase the stress on the patient and his family, reducing control over the disease and impairing her quality of life. Therefore, the data mining algorithm possibly could be a helpful tool for physicians to help with early diagnosis and prevention of recurrence of cancer before engaging in the surgical process of treating the disease as well as incurring high costs for the patient. Of course, these methods are not definitive in medicine, but they can be helpful. So, the main aim of this study is to help the prediction of the recurrence of breast cancer disease using some data mining algorithms mainly based on neural networks. disease forecasting tools and mechanisms along with physicians' experience, can play a helpful role in the correct diagnosis and treatment choice be it and also a matter of increasing confidence in the accurate diagnosis of the disease both the physician and the

**Table 12. Accuracy of the prediction model for each type of recurrence.**

| Type of recurrence | Accuracy |
|---|---|
| Local recurrence | 0.76 |
| Regional recurrence | 0.66 |
| Distant recurrence | 0.87 |
| Local and Distant recurrence | 0.69 |
| Regional and Distant recurrence | 0.70 |
| No recurrence | 0.88 |

**Table 13. The total amount of disease-free survival in the first to fifth years after the initial treatment.**

| year | The number of cumulative cases of the first recurrence | Percentage of disease-free survival | Percentage of disease-free survival prediction | Percentage of the standard error rate of disease-free survival |
|------|------|------|------|------|
| first | 720 | 80 | 76 | 2 |
| second | 1412 | 61 | 59 | 3 |
| third | 1751 | 52 | 51 | 3 |
| fourth | 1941 | 47 | 34 | 3 |
| fifth | 2009 | 45 | 22 | 3 |

patient favorably increased. On the other hand, how the diagnosis will determine the next steps for the patient. So, if it is of low accuracy, it affects one's survival and it is vital to have excellent and acceptable accuracy of forecasting models.

A model that evaluated by examining different criteria, accuracy, and sensitivity is the best, and the higher the forecast, the more reliable it will be.

The current incidence of breast cancer is high and early detection of primary tumors and the strict use of adjuvant therapy not only result in more prolonged survival but also lengthen the disease-free interval. Thus, increasing the number of patients requiring follow-up.

So, help in predicting the recurrence of breast cancer is essential for many reasons. For instance, in the case of patients who have only one tumor or one the breasts entirely removed, if the likelihood of recurrence of breast cancer is high, it can be predicted before the spread of cancer did uniquely to other parts of the body. Predictive models can be helpful and beneficial in this regard. But It should be noted that in the field of evaluation of medical prediction models, at least two features of the model and sensitivity of the model has to be considered because Considering one of them alone can be misleading.

Besides, special attention should be paid to the value of false-negative. It is essential because the patient is mistakenly considered healthy and it can have hazardous consequences.

for limitation in our dataset (we had only 16 complete information about patients that their relapse occurred after 10 years), we have analyzed only patients with shorter than 5 years.

Artificial neural networks are modern disease diagnosis methods that have excited the attention of researchers in recent years. Therefore, seven new and conventional data mining algorithms neural networks: MLP, LVQ, Bayesian Neural Network, KPCA-SVM, C5.0, and Random forest have been used. This study clearly shows the effect of neural networks technology in the recurrence of breast cancer classification. Artificial neural networks can be used as a diagnostic method with high sensitivity and specificity to detect prediction of recurrence of breast cancer tumors, besides other non-invasive diagnostic methods (such as mammography and radiography). The importance of this result is that maybe it can help to prevent patients, who do not need any invasive diagnostic methods (sampling and surgery), from such operations. In addition to the decrease in the costs, faster and more precise diagnostic of prediction of recurrence of breast cancer may increase the treatment chance.

**Table 14. The most important feature with the highest accuracy in the diagnosis of breast cancer.**

| Step No | Feature | Accuracy | Sensitivity | Feature | Time |
|---------|---------|----------|-------------|---------|------|
| 1 | {LN involvement rate} | 95.97 | 90.53 | 98.88 | **53.26** |
| 2 | {LN involvement rate, Her2 value} | 97.80 | 97.89 | 97.75 | **53.18** |
| 3 | {LN involvement rate, Her2 value, Tumor size} | 98.11 | 96.84 | 99.44 | **49.15** |
| 4 | {LN involvement rate, Her2 value, Tumor size, free or closed tumor margin} | 98.24 | 97.89 | 99.44 | **44.78** |
| 5 | {LN involvement rate, Her2 value, Tumor size, free or closed tumor margin, Tumor grade} | 98.24 | 97.89 | 99.44 | **44.78** |

Results show that the C5.0 and the KPCA-SVM have shown better performance in terms of accuracy, true negative, Sensitivity, Specificity, Geometric mean of sensitivity and specificity, PPV, NPV, Geometric mean of PPV and NPV, F-measure, Area under ROC curve in comparison to other methods.

Except for MLP, Bayesian, and LVQ, in terms of area under the ROC curve, the performance of methods is appropriate (more than 0.7). In terms of accuracy, the C5.0 and KPCA-SVM outperformed all other methods; however, other methods Except MLP, Bayesian, and LVQ, also achieved high efficiency (more than 0.70).

Sensitivity ranged from a minimum of 0.509 (MLP) to a maximum of 0.869 (C5.0). Considering that having Recurrence of Breast Cancer is the critical prediction in this biomedical application, a method with higher sensitivity is desired; therefore, methods such as MLP are inappropriate for prediction of Recurrence of Breast Cancer. However, C5.0 showed the best sensitivity result.

Given the nature of the present data and generally, in the field of breast health and cancer research, among the reasons for choosing the decision tree method for prediction, it can be mentioned for breast cancer recurrence data behavior, the decision tree is capable of working with continuous and discrete data (Other methods can only work with one type). Unnecessary comparisons are eliminated in this structure, different properties are used for different samples, no estimation of the distribution function is needed data preparation for a decision tree is unnecessary or straightforward (other methods often require data normalization or deletion of empty values or the creation of null variables). Decision tree structures are dominant for analyzing big data in a short time, finding unexpected or unknown relationships, also this method can adapt to inadequate data. In conclusion, we may be able to facilitate the prediction of breast cancer recurrence by designing suitable machine learning algorithms. In the current study, we found the C5.0 algorithm possibly, could be a helpful tool for predicting by physicians and health care policymakers in breast cancer recurrence prediction at the stage of distant recurrence and nonrecurrence, especially in the first to third years. And also, LN involvement rate, Her2 value, Tumor size, free or closed tumor margin were found to be the most important features in our dataset. This may results in more sustainable health for the patients and, consequently, a lower psychological, social, and economic burden on society.

## Supporting information

**S1 Data.**
(XLSX)

## Author Contributions

**Conceptualization:** Alireza Mosayebi, Barat Mojaradi, Ali Bonyadi Naeini, Seyed Hamid Khodadad Hosseini.

**Data curation:** Alireza Mosayebi, Barat Mojaradi, Ali Bonyadi Naeini, Seyed Hamid Khodadad Hosseini.

**Formal analysis:** Alireza Mosayebi, Barat Mojaradi, Ali Bonyadi Naeini, Seyed Hamid Khodadad Hosseini.

**Investigation:** Alireza Mosayebi, Barat Mojaradi, Ali Bonyadi Naeini, Seyed Hamid Khodadad Hosseini.

**Methodology:** Alireza Mosayebi, Barat Mojaradi, Ali Bonyadi Naeini, Seyed Hamid Khodadad Hosseini.

**Project administration:** Alireza Mosayebi, Barat Mojaradi, Seyed Hamid Khodadad Hosseini.

**Resources:** Alireza Mosayebi.

**Software:** Alireza Mosayebi, Barat Mojaradi, Ali Bonyadi Naeini, Seyed Hamid Khodadad Hosseini.

**Supervision:** Barat Mojaradi, Seyed Hamid Khodadad Hosseini.

**Validation:** Alireza Mosayebi, Barat Mojaradi, Ali Bonyadi Naeini, Seyed Hamid Khodadad Hosseini.

**Visualization:** Alireza Mosayebi, Barat Mojaradi, Seyed Hamid Khodadad Hosseini.

**Writing – original draft:** Alireza Mosayebi, Barat Mojaradi, Ali Bonyadi Naeini, Seyed Hamid Khodadad Hosseini.

**Writing – review & editing:** Alireza Mosayebi, Barat Mojaradi, Ali Bonyadi Naeini, Seyed Hamid Khodadad Hosseini.

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
