## [Decision Letter · Decision Letter 0]

7 May 2020

PONE-D-20-07947

Modeling and Comparing Data Mining Algorithms for Prediction of Recurrence of Breast Cancer

PLOS ONE

Dear Dr Mojaradi,

Thank you for submitting your manuscript to PLOS ONE. After careful consideration, we feel that it has merit but does not fully meet PLOS ONE’s publication criteria as it currently stands. Therefore, we invite you to submit a revised version of the manuscript that addresses the points raised during the review process.

One reviewer argues that more work would be necessary to make the claim that the results are clinically relevant.  This could be addressed by a major reworking of the analysis or by removing or sufficiently qualifying any claims of applicability to clinical or policy decisions.

We would appreciate receiving your revised manuscript by Jun 21 2020 11:59PM. To enhance the reproducibility of your results, we recommend that if applicable you deposit your laboratory protocols in protocols.io, where a protocol can be assigned its own identifier (DOI) such that it can be cited independently in the future. For instructions see: http://journals.plos.org/plosone/s/submission-guidelines#loc-laboratory-protocols

We look forward to receiving your revised manuscript.

Kind regards,

Bryan C Daniels

Academic Editor

PLOS ONE

3. In ethics statement in the manuscript and in the online submission form, please provide additional information about the database used in your retrospective study. Specifically, please ensure that you have discussed whether all data were fully anonymized before you accessed them and/or whether the IRB or ethics committee waived the requirement for informed consent. If patients provided informed written consent to have their data used in research, please include this information.

4. Your ethics statement must appear in the Methods section of your manuscript. If your ethics statement is written in any section besides the Methods, please move it to the Methods section and delete it from any other section. Please also ensure that your ethics statement is included in your manuscript, as the ethics section of your online submission will not be published alongside your manuscript.

6. We note you have included a table to which you do not refer in the text of your manuscript. Please ensure that you refer to Table 1 and 4 in your text; if accepted, production will need this reference to link the reader to the Table.

Reviewers' comments:

Reviewer's Responses to Questions

**Comments to the Author**

1. Is the manuscript technically sound, and do the data support the conclusions?

Reviewer #1: No

Reviewer #2: Yes

2. Has the statistical analysis been performed appropriately and rigorously? 

Reviewer #1: I Don't Know

Reviewer #2: Yes

3. Have the authors made all data underlying the findings in their manuscript fully available?

Reviewer #1: No

Reviewer #2: Yes

4. Is the manuscript presented in an intelligible fashion and written in standard English?

Reviewer #1: Yes

Reviewer #2: Yes

5. Review Comments to the Author

Reviewer #1: Dear authors,

First, I'd like to aknowledge the effort made to improve the manuscript by moving your analysis to a meaninful clinical question.

Again, my main concern is that the interpretation of the clinical scenario is not correct, hence the conclusion "Therefore, machine learning algorithms, in particular, the C5.0, can be of great help to physicians and health care Policy Makers, especially in predicting recurrence of breast cancer." raised from your analysis is not supported.

There are many things regarding the experimental setup that should be amended.

First of all: recurrence in breast cancer is a time dependent event. This has been prevously established without doubt. I think it's mandatory to include time to recurrence in order to predict this recurrence.

Second: Its also been stablished that, for brast cancer recurrence studies a minimun follow up of 5 years its necessary to have a complete picture of the recurrences. This is due to the huge heterogeneity of breast cancer. For example, most TNBCs will relapse in the first three years, while ER/PR+ breast cancer recurrence is spread among the first 10 years. This can bias the analises because most of the recurrences will be from TNBCs.

Third: I strongly recommned including a clinical advisor in this work. There are some variable interpretations that should be corrected to faithfully reflect the clinical scenario (sfor examples: ER/PR are analized independently but no together, as in the clinical practice). No data about the number of recurrences in this population is presented. Obviously, treatment is related with relapse. But it should not be included as a variable to predict relapse, because the objective is to be able to predict this relapse prior to treatment in ER/PR+ breast cancer, allowing to decide fi chemotherapy is needed or not. On the other hand, prediction of relapse in TNBC will follow other clinical objectives (find out chemotherapy resistant tumors to provide these patients with additional tratment options via clinical trials for example. All these differences in the clinical interpretation should be well defined prior to the analysis, and must conditionate the analysis itself to demonstrate the capability of this powerful mathematical tools in the clinical setting.

And finally, I want to raise a question. It is not possible, with the large number of patients included, to split them onto training and test? this could help to estimate the overfitting of the methods.

Reviewer #2: Early detection of recurrence of Breast cancer can provide potential advantage in the treatment of this disease. There have been many researches in the recent past about finding the most critical attributes that plays a major role in prediction of recurrence of breast cancer. However, in this research, the author has also interviewed with specialists in the field of breast cancer along with data mining techniques. Thus the authenticity of this work increases.

Since it is a revised paper thus below are my reviews about the paper based on the previous review comments are as follows.

Reviewer 1 Comments

Comment 1 and Comment 2:

The author has resolved this issue. As per the comment of reviewer 1, the author has used 7 data mining methods and predictions has been now on finding the recurrence of breast cancer. The attributes have been now described.

Reviewer 2 Comments

Comment 1 and Comment 2:

The author has resolved this issue. As per the comment of reviewer 2, the author has used new prediction algorithms and to evaluate performance F1 and area under ROC curve is also used.

As per the revised paper, the author has modified the paper according to the reviewer comments.

6. PLOS authors have the option to publish the peer review history of their article (what does this mean?). If published, this will include your full peer review and any attached files.

Reviewer #1: Yes: Angelo Gámez-Pozo

Reviewer #2: No

---

## [Author Response · Author response to Decision Letter 0]

10 Jul 2020

Dear Editor-in-Chief,

We wish to thank you, the associate editor, and the reviewers for the comments we received on the attached paper, and also thank you for allowing us to revise the manuscript again which has helped to enhance the quality of the paper. We hereby submit a revised version of the paper with an ID number of PONE-D-20-07947. Significant changes and modifications were highlighted in the revised manuscript, and the detailed responses to reviewers’ comments are listed as follows. We hope that the modified version is acceptable, and we look forward to your kind recommendations.

Best regards,

The Authors

Editor Comments to the Author

Comment 1:

One reviewer argues that more work would be necessary to make the claim that the results are clinically relevant. This could be addressed by a major reworking of the analysis or by removing or sufficiently qualifying any claims of applicability to clinical or policy decisions.

Response and corrections regarding comment 1:

According to your helpful comment, the manuscript is carefully reviewed, some sentences for eliminating and decreasing ambiguity of the results and conclusion of our analysis are made more clearer and some sentences removed. Moreover, instead of definite statements, we have used possibly, could be a helpful and helpful approach to conclusions in the diagnosis phase of clinical scenarios. it is stated that from the clinical scenario stages, our research possibly, could be helpful for physicians in the detection stage with a good probability, before entering the costly and stressful treatment(please see section “Abstract” in page 1 , and page 3 in section “introduction”, lines 82-88 and page 21-22 in section “Conclusion and Discussion”). Also, some sentences for eliminating ambiguity and make the results of the analysis clearer, are removed and some sentences placed in the methodology section (please see page 3 in the section "Materials and Methods", lines 106-111, and pages 8-10 in section "Results”). more patients' clinical characteristics are added (please see page 8 in section “Results”, lines 249- 250 and page 9. Line 251-253). More tables and results from our analysis are added to the end of results to sufficiently qualifying the medical output at each recurrence stage (please see page 18-21 in section “Results”). The most important features are obtained by two methods: chi-square (on page 8) and forward feature selection method (please see section “Feature importance analysis" on page 17 and section “Important features for breast cancer metastasis prediction” in pages 20-21). 

Review Comments to the Author:

Reviewer 1 Comments:

First, I'd like to acknowledge the effort made to improve the manuscript by moving your analysis to a meaningful clinical question. Again, 

Comment 1:

 My main concern is that the interpretation of the clinical scenario is not correct, hence the conclusion "Therefore, machine learning algorithms, in particular, the C5.0, can be of great help to physicians and health care Policy Makers, especially in predicting recurrence of breast cancer." raised from your analysis is not supported.

Response and corrections regarding comment 1:

According to your helpful comment, the manuscript is carefully reviewed, some sentences for eliminating and decreasing ambiguity of the results and conclusion of our analysis are made more clearer and some sentences removed. Moreover, instead of definite statements, we have used possibly, could be a helpful and helpful approach to conclusions in the diagnosis phase of clinical scenarios. it is stated that from the clinical scenario stages, our research possibly, could be helpful for physicians in the detection stage with a good probability, before entering the costly and stressful treatment(please see section “Abstract” in page 1 , and page 3 in section “introduction”, lines 82-88 and page 21-22 in section “Conclusion and Discussion”). Also, some sentences for eliminating ambiguity and make the results of the analysis clearer, are removed and some sentences placed in the methodology section (please see page 3 in the section "Materials and Methods", lines 106-111, and pages 8-10 in section "Results”). more patients' clinical characteristics are added (please see page 8 in section “Results”, lines 249- 250 and page 9. Line 251-253). More tables and results from our analysis are added to the end of results to sufficiently qualifying the medical output at each recurrence stage (please see page 18-21 in section “Results”). The most important features are obtained by two methods: chi-square (on page 8) and forward feature selection method (please see section “Feature importance analysis" on page 17 and section “Important features for breast cancer metastasis prediction” in pages 20-21). 

Comment 2:

There are many things regarding the experimental setup that should be amended.

recurrence in breast cancer is a time-dependent event. This has been previously established without doubt. I think it's mandatory to include time to recurrence to predict this recurrence. 

Response and corrections regarding comment 2:

Based on this helpful and valuable comment, the manuscript has revised and clarified that the present study is performed based on the cross-sectional method and we have used Data Mining algorithms for Cross-Sectional dataset which examines the relationship between disease (or other health-related features) and other variables of interest as they exist in a defined population at a single point in time or over a short period. (please see section “Materials and Methods”, line 106-111 on page 3).

Comment 3:

Its also been established that for breast cancer recurrence studies a minimum follow up of 5 years its necessary to have a complete picture of the recurrences. This is due to the huge heterogeneity of breast cancer. For example, most TNBCs will relapse in the first three years, while ER/PR+ breast cancer recurrence is spread among the first 10 years. This can bias the analysis because most of the recurrences will be from TNBCs.

Response and corrections regarding comment 3:

Thanks for your valuable comment. As mentioned in the reply to the previous comment, based on disease nature and our some limitations in the dataset for analyzing with more accuracy in machine learning algorithms and improve conclusions, we have used a cross-sectional statistical method. According to your valuable and precise comment, we have revised and clarified that data gathering of this research performed from June 2018 to June 2019 the official statistics of the Ministry of Health and Medical Education and the Iran Cancer Research Center for patients with breast cancer who had been followed for a minimum of 5 years from February 2014 to April 2019, including 5471 independent records. Based on some studies about breast cancer recurrence for example Wangchinda & Ithimakin (2016), breast cancer relapse occur and analyze generally in two periods (shorter than 5 years or longer than 5 years). But for limitation in our dataset(we had only 16 complete information a bout patients that their relapse occurred after 10 years), we have analyzed only patients with shorter than 5 years and we have added some analysis a bout this first five years(please see section “Materials and Methods”, line 106-111 on page 3 and table 12 in page 19 of section “results”).

 And we think in this stage, it possibly can be valuable to predict the recurrence of the disease in shorter periods because most relapses occur during the first 5 years after diagnosis. By defining algorithms, it possibly could be helpful to predict the disease recurrence in a shorter period, help to physicians for detection and management of recurrence.

 However, according to your helpful comment, we absolutely try to gather more information about our dataset and conduct more practical analyze, in the future of our studies for patients with breast cancer recurrence in longer time to improve our results and conclusions.

Comment 4:

I strongly recommend including a clinical advisor in this work. There are some variable interpretations that should be corrected to faithfully reflect the clinical scenario (for example ER/PR are analyzed independently but no together, as in the clinical practice). No data about the number of recurrences in this population is presented. Obviously, treatment is related with relapse. But it should not be included as a variable to predict relapse, because the objective is to be able to predict this relapse prior to treatment in ER/PR+ breast cancer, allowing to decide if chemotherapy is needed or not. On the other hand, prediction of relapse in TNBC will follow other clinical objectives (find out chemotherapy resistant tumors to provide these patients with additional treatment options via clinical trials for example. All these differences in the clinical interpretation should be well defined prior to the analysis, and must conditionate the analysis itself to demonstrate the capability of this powerful mathematical tools in the clinical setting.

Response and corrections regarding comment 4:

Based on your helpful and valuable comment, the manuscript has reviewed and revised by more clinical advisors. At the first stage of our research, we have concluded from some studies for example Iqbal & Buch(2016) , Lim, Palmieri, & Tilley(2016) and Patani & Martin(2014) and with confirmation of physicians related to our dataset patients, that about 80% of breast cancers are “ER-positive”, which means the cancer cells grow in response to the hormone estrogen. About 65% of these are also “PR-positive.” They grow in response to another hormone, i.e., progesterone. hence, these two criteria can be analyzed independently. 

But In this study breast cancer is classified into four groups based on IHC profile ER/PR and Her2/neu expression, positive (+), and/or negative (−). The groups are:

ER/PR+, Her2+ = ER+/PR+, Her2+; ER−/PR+, Her2+; ER+/PR−, Her2+

ER/PR+, Her2− = ER+/PR+, Her2−; ER−/PR+, Her2−; ER+/PR−, Her2−

ER/PR−, Her2+ = ER−/PR−, Her2+

ER/PR−, Her2− = ER−/PR−, Her2−

And are not considered independently. The IHC classification correlates well with intrinsic gene expression microarray categorization: ER/PR+, Her2+ with Luminal B; ER/PR+, Her2− with Luminal A; ER/PR−, Her2+, and ER/PR−, Her2− with triple-negative/basal-like tumors. Apart from lending itself to subtype analyses of tumors when fresh tissue is not available, the IHC classification has prognostic and therapeutic implications, is inexpensive and readily available. In general, we paid attention to the relevant factors and separate factors, but some explanations were not provided to prevent the length of the article content and a large number of pages. But for information limitation in our dataset (we had only 16 complete information about patients that their relapse occurred after 10 years), we have analyzed only patients with shorter than 5 years recurrence. Also, based on most of studies, the greatest risk of recurrence is in the 5 years after breast cancer diagnosis. Based on your valuable opinion, we have added some explanations to make the explanations more clear (please see line 212-222, on page 6 of section “results”). 

Also, of course, your concern about the dependent variable is valuable, precise, and helpful for us to revise and clarify our explanations about input features to machine learning algorithms. at the first stage, based on chi-square results in table 5, we have extracted most related factors to breast cancer recurrence, then, after refining the results and discarding the treatment and diagnosis methods, 3 factors are considered as the most important factors affecting the recurrence of breast cancer in our dataset by a chi-square method(please see page 8-10 in section “Results”). This is one of the two methods that we have used to extract important features. Another method is forward feature selection in machine learning. We have used these two methods for validating the results of the best features extraction analysis (please see section “Feature importance analysis" on page 17 and section “Important features for breast cancer metastasis prediction” on page 20-21).

But we clarified that to the prediction of breast cancer recurrence, by discarding 6 factors of treatment and diagnosis, other influencing features are considered as the inputs of machine learning algorithms (please see page 10 in section “Results”, lines 263-268).

Furthermore, more patients' clinical characteristics and information have added (please see page 8 in section “Results”, lines 249- 250 and page 9, line 251-253 and Tables 10, 11 and 13 in pages 18-19). For validation, we have compared the results of the machine learning algorithms with the clinical results of the patients' dataset that we have. Tables 10-13 on pages 18-19 represent this compare and analysis (please see section “Results”, Tables 10-13 on page 18-19). 

In the present study, we tried to collect a large number of patients data with some limitations in clinical information, and due to the condition and limitation we faced and consulted with physicians related to patients, we tried to analyze dataset to help diagnose of cancer recurrence in the early stages and to use machine learning as a mathematical tool in the clinical stages. However, according to your helpful comments, we absolutely try to gather more information about our dataset and conduct more practical analyze, in the future of our studies.

Comment 5:

finally, I want to raise a question. It is not possible, with a large number of patients included, to split them onto training and test? this could help to estimate the overfitting of the methods.

Response and corrections regarding comment 5:

Thanks for this valuable question. Overfitting happens when the learning algorithm continues to develop hypotheses that reduce training set error at the cost of an increased test set error. There are several approaches to avoiding overfitting. In this research we have used a nested 5-fold cross-validation approach to train (four folds) and test (one fold) the models. Patients meeting the inclusion criteria are randomly assigned to one of the five outer folds. To ensure that the important feature set was generated from real patients with breast cancer and the importance of the features was not emphasized by duplicating minor cases, we chose the under-sampling approach to build the model. We randomly selected 20 sets of controls in each round of cross-validation, matching the number of cases, and generated 80 training datasets by using one set of controls and all cases. In each training step, we used 5-fold inner cross-validation to tune the models. this method has used but for preventing of exceeding machine learning analysis we have not included in our manuscript with so explanation and only we mentioned the final result of this result in our paper, but according to your valuable concern, we have added these explanations in our manuscript(please see page 4-5 in the section “material and methods’).

Finally, the authors are thankful to the editors and reviewers for the critical comments and constructive suggestions, which helped improve the quality and presentation of the paper significantly.

references:

1. Wangchinda, P., & Ithimakin, S. (2016). Factors that predict recurrence later than 5 years after initial treatment in operable breast cancer. World Journal of Surgical Oncology, 14(1), 223. doi:10.1186/s12957-016-0988-0

2. Iqbal, B., & Buch, A. (2016). Hormone receptor (ER, PR, HER2/neu) status and proliferation index marker (Ki-67) in breast cancers: Their once-pathological correlation, shortcomings, and future trends. Medical Journal of Dr. D.Y. Patil University, 9(6), 674-679. DOI:10.4103/0975-2870.194180

 3. Lim, E., Palmieri, C., & Tilley, W. D. (2016). Renewed interest in the progesterone receptor in breast cancer. British Journal of Cancer, 115(8), 909-911. DOI:10.1038/bjc.2016.303

4. Patani, N., & Martin, L. A. (2014). Understanding response and resistance to estrogen deprivation in ER-positive 

breast cancer. Molecular and cellular endocrinology, 382(1), 683-694. DOI:10.1016/j.mce.2013.09.038

---

## [Editor Report · Decision Letter 1]

31 Jul 2020

Modeling and Comparing Data Mining Algorithms for Prediction of Recurrence of Breast Cancer

PONE-D-20-07947R1

Dear Dr. Mojaradi,

We’re pleased to inform you that your manuscript has been judged scientifically suitable for publication and will be formally accepted for publication once it meets all outstanding technical requirements.

Note that there are also grammatical and copyediting issues that will need to be addressed before publication.

Kind regards,

Bryan C Daniels

Academic Editor

PLOS ONE
---

## [Editor Report · Acceptance letter]

27 Aug 2020

PONE-D-20-07947R1 

Modeling and Comparing Data Mining Algorithms for Prediction of Recurrence of Breast Cancer 

Dear Dr. Mojaradi:

I'm pleased to inform you that your manuscript has been deemed suitable for publication in PLOS ONE. Congratulations! Your manuscript is now with our production department. 

Kind regards, 

on behalf of

Dr. Bryan C Daniels 

Academic Editor

PLOS ONE